# Assessment of Spatial-Temporal Changes of Landscape Ecological Risk in Xishuangbanna, China from 1990 to 2019

Yun Liu [1], Weiheng Xu [1,*], Zehu Hong [1], Leiguang Wang [2], Guanglong Ou [3] and Ning Lu [1]

1. College of Big Data and Intelligent Engineering, Southwest Forestry University, Kunming 650233, China
2. Institute of Big Data and Artificial Intelligence, Southwest Forestry University, Kunming 650233, China
3. College of Forestry, Southwest Forestry University, Kunming 650233, China
* Correspondence: xwh@swfu.edu.cn; Tel.: +86-388-883-2080

**Abstract:** Xishuangbanna is a major natural rubber and tea production base in China and a national nature reserve with the best-preserved tropical ecosystem. However, the extensive exploitation and use of land resources impact the land use/land cover (LULC) and the processes of regional landscape ecology, further causing a battery of ecological and environmental problems. It is necessary to evaluate landscape ecological risk objectively and quantitatively for improving the ecological environment and maintaining ecological balance. First, this study selected China Land Cover Dataset (CLCD) to analyze the changes in LULC. Second, we constructed the landscape ecological risk index (ERI) using LULC changes based on Google Earth Engine (GEE) platform. Third, the spatial-temporal pattern and spatial autocorrelation of landscape ecological risk were assessed in our study area. The results showed that the significant change in LULC was that the areas of cropland increased, and the areas of forests decreased during 1990–2019; the forests of a total area of 859.93 km$^2$ were transferred to croplands. The landscape ecological risk kept a low and stable level from 1990 to 2019, more than 75% of the study area remained at the lower or lowest risk level, and in about 70% of the total study area, the ERI level maintained stability. In addition, the landscape ecological risk of the Xishuangbanna increased during 1990–2010 and decreased during 2010–2019. The ecological risk was a significant spatial autocorrelation and has been an aggregation trend in space from 1990 to 2019. Our research can identify key risk areas and provide a reference for the management and sustainable use of land resources, which promotes the understanding of landscape ecological risk and sustainable development of the ecological environment.

**Keywords:** landscape ecological risk; Xishuangbanna; land use/land cover; spatial-temporal pattern; spatial auto-correlation analysis



## 1. Introduction

With the acceleration of the world's industrialization process, global ecological problems have become increasingly prominent [1]. The realization of ecologically sustainable development has become a common target for all human beings. Due to the growth of the population and the development of economics in the past decades, the change rate of land use/land cover (LULC) accelerated, and the land use intensity was an increasing trend in Xishuangbanna [2]. Li et al. investigated dynamic change characteristics of LULC in Xishuangbanna from 1996 to 2016 [3], while the forest area decreased by 297.21 km$^2$, but the rubber area increased by 537.93 km$^2$. After, Cao et al. [4] found that the rubber area in Xishuangbanna increased by about 3500 km$^2$ during 1976–2015, yet forests decreased by about 2500 km$^2$. The building area increased 12 times and reached more than 190 km$^2$, with most of the increase shifting from shrub and paddy fields. In addition, the fact that the rubber plantation area increased by 33.53% in Menglun County, Xishuangbanna, from 1988 to 2006, while forest and cropland decreased by 21.16% and 12.68%, was discovered by Hu et al. [5]. However, the LULC shifted from tropical forests of ecological importance and

traditionally managed cropland to large rubber plantations and tea plantations, increasing the deterioration of the ecological environment in Xishuangbanna [6,7]. According to the research of some scholars, it is found that the large-scale change in LULC caused an imbalance in the internal structure and function of the natural ecological system [8]. Large-scale land development and deforestation inevitably caused some ecological problems, such as soil denudation, sandy desertification, and biodiversity loss [9].

In general, these ecological and environmental problems changed the landscape ecological risk of the study area, and landscape ecological risk was the most excellent in areas with a high frequency of LULC change [10]. In turn, the increased ecological risk restricts the sustainable development of land, which creates a vicious cycle [11]. Therefore, the ecological environment quality degraded in Xishuangbanna with the expansion of rubber plantations increased during 1995–2007 [12]. Furthermore, the diversion of large-scale forests and cropland to rubber plantations and cities has caused a variety of environmental or ecological problems including landscape fragmentation, biodiversity loss, and environmental pollution in Xishuangbanna [3]. Moreover, Liu et al. [13] discovered that the change range of ecological risk caused by LULC change was different at the global and local levels. Local ecological risk usually increased when the area of nearby construction land increased. Zhang et al. [14] maintained that the changing pattern of LULC type corresponds to the changing patterns of ecological risk type. Ecological degradation occurs when high-risk types of land are vulnerable to environmental and human disturbances dominated in the area, such as cropland, construction, and water. Consequently, Wang et al. [15] presented that the higher and highest risks were primarily distributed in the experimental area dominated by cropland, and the nature reserve plays a significant role in maintaining the sustainable development of the ecosystem. The assessment of landscape ecological risk using LULC changes based on a spatial pattern perspective was a regional ecological risk assessment method [16]. This method can identify and predict the impact of LULC change caused by human activities on landscape components, construction, function, and procedure in each region [9]. With the development of ecological risk research, human disturbance caused by land-use change has become a research hotspot in ecological risk assessment (ERA) [17,18].

Risk assessment was proposed first in the 1980s for human health assessment. In the following years, the ecological effects of chemical pollutants were shown by environmental impact assessments [19]. Since the 1990s, gradually turning the field of risk assessment to ERA, the risk factors and receptors trend towards more diversification, and a relatively perfect ERA system has been initially formed [20]. Since the landscape ERA reflects typical spatial variability and scale dependency, it is often used to evaluate the likelihood and extent of adverse impacts of human activities or changes in natural factors on ecosystems [9,15]. In recent years, the landscape ERA mainly adopts risk source [21,22] and landscape pattern [10,23] as two evaluation methods. The landscape ERA mainly focuses on the impact of LULC change based on landscape patterns. Landscape patterns can quantitatively reflect LULC change [10]. However, the focus of risk source assessment is the risk effect of the existing landscape pattern deviating from the optimal model, not a specific and clear disturbance source [24,25]. In addition, the ecologically fragile areas are a research hotspot in existing research fields of landscape ecological risk assessment, such as industrial and mining areas [18], watersheds [6,10,26], nature reserves [15,27], and large cities [9,28,29]. Moreover, spatial autocorrelation analysis can explore the agglomeration law of landscape ecological risk, can effectively help highlight the spatial pattern of ecological risk distribution, and can quickly analyze the high-risk areas that need to be concerned in the study area [30,31]. However, as one of the national nature reserves in China, Xishuangbanna lacks a landscape ecological risk assessment. Meanwhile, the traditional professional remote sensing software applied to assess long-term changes in the landscape ecological risk assessment is complicated and time-consuming, such as ArcGIS and Fragstats [32].

Google Earth Engine (GEE) is a free and open-access platform [33], with a powerful capability in accessing and directly processing multi-temporal, multi-sources, multi-scale, and large-scale scope data [34,35]. Additionally, it has been utilized widely in areas of ecological assessment [36], especially in large-scale scope and long-term changes [12,26]. Moreover, the platform is also built in the Application Programming Interface(API) of JavaScript and Python to help scientists develop their algorithms [37]. Therefore, compared to the traditional software, the GEE has more potential for landscape ecological risk assessment on a long-term and large scale [38]. However, few studies have applied the GEE platform to the areas of landscape ecological risk assessment.

As a national nature reserve of tropical rainforest and the second largest natural rubber production base and the origin of Pu'er tea in China, Xishuangbanna has changed the original structure of LULC due to the establishment of large-scale rubber plantations and the implementation of nationally coordinated policies [39]. However, the landscape ecological risk is affected by LULC changes [10]. Therefore, the assessment of the spatial and temporal pattern of landscape ecological risk is grounded in LULC changes over the past 30 years, which is indispensable to promoting sustainable development in the social economy and the ecological system of Xishuangbanna. The objectives of this study were to: (1) to monitor the dynamic changing characteristics of LULC from 1990 to 2019 in Xishuangbanna, (2) to explore the spatial and temporal changes of landscape ecological risk by using LULC change in Xishuangbanna, and (3) to discuss the spatial autocorrelation of ecological risk at the landscape level.

## 2. Materials and methods

### 2.1. Study Area

Xishuangbanna Dai Autonomous Prefecture (21°08′–22°36′ N, 99°56′–101°50′ E) consists of three administrative counties (Jinghong, Menghai, and Mengla) that are seated in southwestern Yunnan Province, China. Xishuangbanna is approximately 19,150 km$^2$, which is adjacent to Laos and Myanmar in the south and southwest, respectively (Figure 1) [39]. The terrain in the north is high, and in the south is low, and is dominated by hilly mountains, with an elevation between 475 m and 2428 m [27]. The rainy season starts in May and ends in October, and the average annual precipitation is from 1500 mm to 2000 mm [40]. The average annual temperature is above 20 °C.

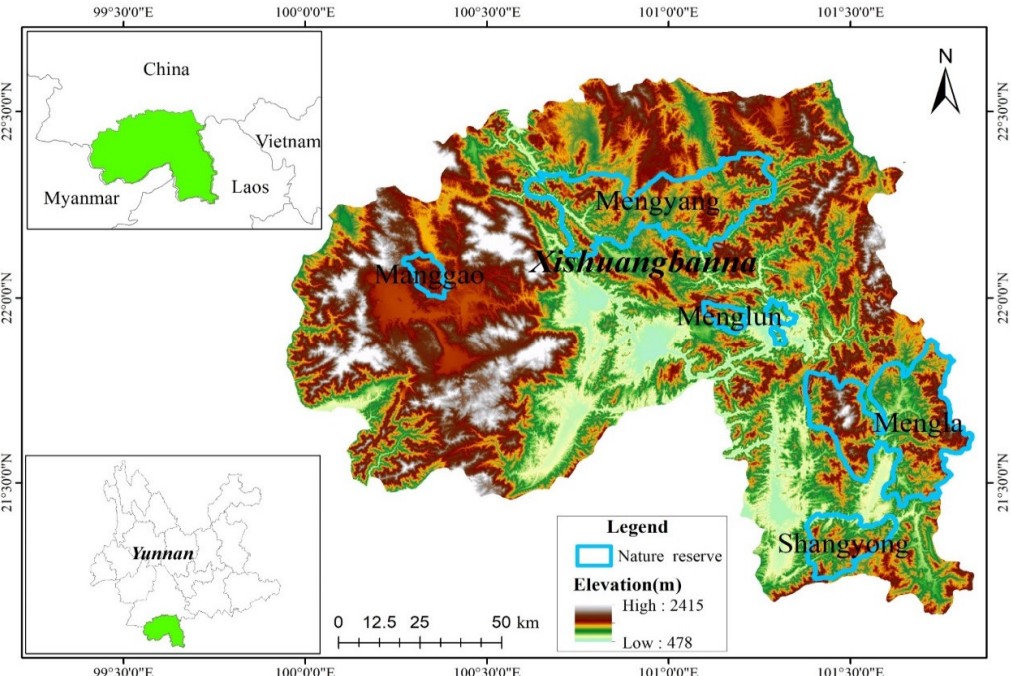

**Figure 1.** Location of Xishuangbanna Prefecture.

The unique tropical rainforest nature reserve (Nature Reserve) in China is located in Xishuangbanna, with an area of 2474 km$^2$, accounting for 12.9% of Xishuangbanna's land area. The Nature Reserve comprised five geographically unconnected sub-nature reserves (Mengyang, Menglun, Mengla, Shangyong, and Manggao) [41]. With the main purpose of protecting tropical forest ecosystems and rare wild animals and plants, the Nature Reserve has the largest tropically virgin forest area in China, with a complete tropical forest ecosystem and abundant biological resources. In addition, 109 species of animals and 56 species of plants are listed as national key protected objects. However, rubber plantations and other human activities caused LULC changes in Xishuangbanna, which affected biodiversity and ecosystem services [42].

*2.2. Data Sources*

We selected the LULC data of Xishuangbanna in 1990, 2000, 2010, and 2019 from the China Land Cover Dataset (CLCD) produced by Huang Xin team of Wuhan University, and the spatial resolution of CLCD data is 30 m, accessed on 11 August 2021 and freely available at (https://doi.org/10.5281/zenodo.4417810) [43]. The CLCD dataset was mainly produced based on the random forest (RF) classification of Landsat images on the GEE platform. This design of classification system is similar to that of finer resolution observation and monitoring of global land cover (FROM_GLC) and can be remapped to the existing United Nations Food and Agriculture Organization (FAO) land-cover classification system as well as the International Geosphere-Biosphere Programme (IGBP) system [44], including nine major land cover classes(LCC): cropland, forest, shrub, grassland, water, snow and ice, barren, impervious, and wetland. The accuracy of the data in each period was verified before being used, and the overall accuracy (OA) of CLCD is 79.30% $\pm$ 1.99%, which is higher than the accuracy of existing land cover (LC) products (i.e., MCD12Q1, ESACCI_LC, FROM_GLC, and GlobeLand30) [43], meeting the research needs herein. Therefore, we selected the classification results of CLCD to reclassify them into six LCC (cropland, forest, shrub, impervious, water, and others), which are used to evaluate the spatial and temporal patterns of landscape ecological risk for Xishuangbanna during 1990–2019.

*2.3. Methodology*

We established a specific flowchart for this study (Figure 2). First, the CLCD dataset was used to analyze the dynamic change of LULC from 1990 to 2019 for Xishuangbanna based on the GEE platform. Second, we produced four 1 km × 1 km landscape ecological risk level maps in 1990, 2000, 2010, and 2019, respectively. Third, we analyzed the spatial-temporal changes in the landscape ecological risk for Xishuangbanna based on landscape ecological risk index maps from 1990 to 2019. Fourthly, the spatial autocorrelation of the ERI is analyzed through global spatial auto-correlation and local indicators of spatial association.

2.3.1. Land Use/Land Cover Transfer Matrix

The land-use transfer matrix could quantitatively reveal the transformation direction and intensity between different land-use types in a certain period [9]. We applied it to analyze the dynamic transformation of LULC in four periods 1990 to 2000, 2000 to 2010, 2010 to 2019, and 1990 to 2019, respectively, and this method helps us reveal more accurately the internal connection between the LULC change and the landscape ecological risk change. The calculation of the transfer matrix is as follows:

$$S_{ij} = \begin{bmatrix} S_{11} & S_{12} & \cdots & S_{1m} \\ S_{21} & S_{22} & \cdots & S_{2m} \\ & \vdots & \vdots & \vdots & \vdots \\ S_{m1} & S_{m2} & \cdots & S_{mm} \end{bmatrix} (i,j = 1,2\ldots\ldots m) \tag{1}$$

where $S_{ij}$ is the area of land type $i$ shifted inland type $j$; $m$ is the number of land-use types.

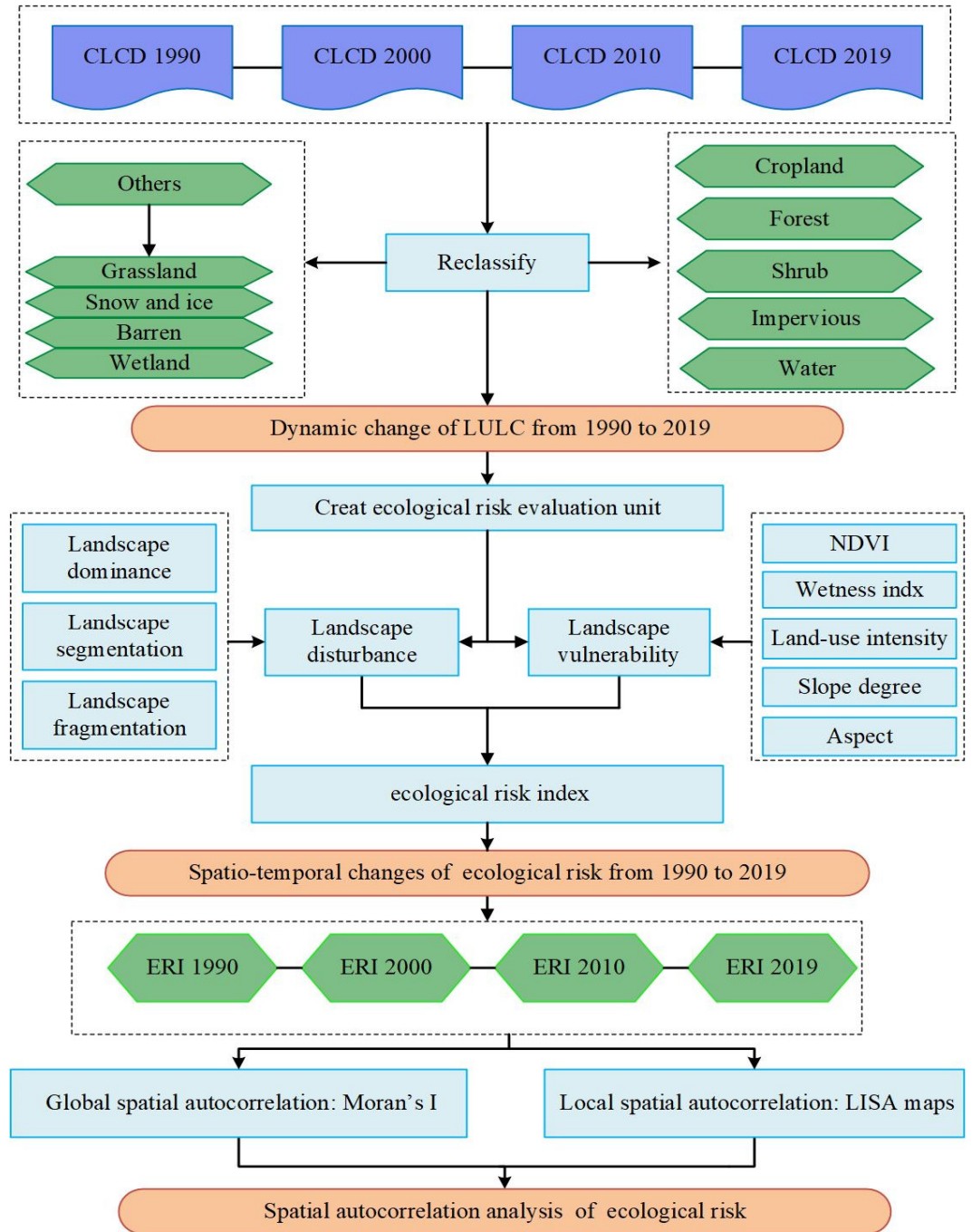

**Figure 2.** Flowchart of assessment of temporal-spatial changes of ecological risk in long time series.

### 2.3.2. Construction of Landscape Ecological Risk Evaluation Model
Determine the Risk Evaluation Cell

Landscape cell size is considered to be 2~5 times the average area of the patch, which is based on landscape ecology advice and landscape spatial heterogeneity, patch size, and watershed area [45]. The average patch size in Xishuangbanna was 0.15 km². To analyze and calculate the landscape ecological risk, based on the actual condition and previous research [9], Xishuangbanna was divided into 21,214 ecological risk assessment cells by the equal interval sampling method, with 1 km × 1 km each cell. In addition, we calculated the ERI for one grid then the results were assigned to the central point of the assessment grids.

Construction of Landscape Ecological Risk Index (ERI)

The ERI is closely related to the landscape patterns and ecological risk; therefore, the ERI is specifically used to reflect their relationships based on LULC changes [28]. Therefore, we constructed the ERI in 1990, 2000, 2010, and 2019 to explore spatial distribution characteristics and temporal change trends in landscape ecological risk for Xishuangbanna. The formulas are as follows:

$$ERI_n = \sum_{i=1}^{m} \frac{A_{ni}}{A_n} \times R_i \tag{2}$$

where $ERI_n$ represents the ecological risk index of the $n$th risk cell; $A_{ni}$ represents the total area (km$^2$) of the $i$th landscape type in the $n$th landscape cell; $A_n$ represents the total area of the $n$th landscape cell; $m$ represents the number of different landscape classes in this study, $m = 6$. The landscape loss index ($R_i$) can reflect the relationship between landscape ecological risk and LULC structure [11], which is expressed as follows:

$$R_i = U_i \times F_i \tag{3}$$

where $R_i$ indicated the ecological loss of landscape type $i$ when experiencing disturbance. The landscape vulnerability index ($F_i$) [46] and landscape disturbance index ($U_i$) [14] are selected to construct the $R_i$ of landscape type $i$. $F_i$ and $U_i$ are shown in Equations (4) and (5), respectively.

$$F_i = f(NDVI, Wet, Land\_use, Slope, Aspect) \tag{4}$$

where $F_i$ reflects the ecological sensitivity of landscape type $i$ to disturbance by stress factors; $NDVI$ and $Wet$ represent the greenness and wetness, respectively; $Land\_use$ represents efficiency of land resource use; $Slope$ represents degree of slope; and $Aspect$ represents slope direction. In this paper, we first normalize the value of five indicators to between 0 and 1, since respective indicator has an individual unit and number range, and then the five-component indicators mentioned above were coupled by principal component analysis (PCA), using the first principal component (PC1) to build the $F_i$. [23]. Finally, $F_i$ values of the six landscape types in our study are: 1 for the impervious, 2 for the forest, 3 for shrub, 4 for others, 5 for cropland, and 6 for water.

$$U_i = aC_i + bS_i + cD_i \tag{5}$$

where $U_i$ represents the landscape disturbance degree index of $i$th landscape type; $C_i$ represents the fragmentation of landscape; $S_i$ represents the isolation of landscape; $D_i$ is the dominance index of landscape; $a$, $b$, and $c$ represent the weights of $C_i$, $S_i$, and $D_i$, respectively, reflecting the effect of human disturbances on landscape ecosystem; and $a + b + c = 1$. According to the previous studies [10,14,15,29], the $C_i$ is considered the most important factor, followed by $S_i$ and $D_i$. Therefore, the weight value of $a$, $b$, and $c$ were 0.5, 0.3, and 0.2, respectively. The formula of $C_i$, $S_i$ and $D_i$ are expressed as:

$$C_i = \frac{m_i}{A_i} \tag{6}$$

$$S_i = \frac{A}{2A_i} \times \sqrt{\frac{m_i}{A}} \tag{7}$$

$$D_i = \frac{(Q_i + M_i + L_i)}{3} \tag{8}$$

where, $m_i$ is the number of patches of landscape type $i$; $A_i$ is the total area of landscape type $i$; $A$ is the total landscape area of the study area; $Q_i$ is the ratio of the landscape $i$ area to the total area of the study area; $M_i$ is the number of patches of landscape $i$ divided by the number of total patches; $L_i$ is the ratio of the number of evaluation units that the landscape $i$ appears to the total number of evaluation units [15].

The greater the ERI value, the higher the landscape ecological risk level of the evaluation cell [10,15]. The ERI was divided into five levels by the natural break point method [9]:

Lowest risk [0–0.03], Lower risk [0.03–0.1], Middle risk [0.1–0.2], Higher risk [0.2–0.4], and Highest risk [0.4–1].

2.3.3. Spatial Autocorrelation Analysis Methods

The spatial correlation of variables can be quantitatively reflected in spatial statistical analysis, which is often used to study regional land-use change and spatial pattern characteristics [47]. Spatial autocorrelation is an important index and is especially used to test the correlation between the ecological risk of the space reference cells and its adjacent spatial ecological risk [26,30]. The homogeneity of the spatial distribution of landscape ecological risk in the study area can be used to describe the spatial correlation of the landscape ecological risk index. The global univariate spatial auto-correlation (Moran's I) and local univariate indicator of spatial association (LISA) were selected to reflect the spatial distribution of ERI in the Xishuangbanna, which were conducted by GeoDa 1.14 software [48].

The Moran's I reflects the correlation strength of the attributed values of adjacent spatial cells. The absolute value is close to 1, indicating a more positive spatial autocorrelation [31].

$$\text{Moran's I} = \frac{\sum_i^n \sum_j^n W_{ij}(x_i - \bar{x})(x_j - \bar{x})}{S^2 \sum_i^n \sum_j^n W_{ij}(x_{i-}\bar{x})^2} \tag{9}$$

where $x_i$ represents the ERI values of reference cell $i$, and $x_j$ is the ERI of adjacent cell $j$, respectively; $n$ is the number of cells indexed by $i$ and $j$, respectively; $W_{ij}$ represents the weight matrix of the adjacent space; when cell $i$ is neighbor to cell $j$, $W_{ij} = 1$, otherwise, $W_{ij} = 0$; $S^2$ is the mean square deviation ($i = 1, 2, \ldots, n$). Furthermore, Moran's I ranges from minus 1 to 1. The ERI is a positive spatial autocorrelation when the Moran's I values are between 0 and 1, and is a negative spatial correlation when the Moran's I values are from minus 1 to 0, but 0 represented uncorrelated in space [15].

The local indicator of spatial association reflects the local spatial heterogeneity of landscape ecological risk distribution and the spatial differences of assessment unit I and its surroundings [14]. The calculation formula is as follows:

$$\text{LISA} = z_i \sum_j^n W_{ij} z_j (i \neq j) \tag{10}$$

where $z_i$ and $z_j$ represent the standardization of ERI in cell $i$ and cell $j$, respectively; $w_{ij}$ represents the spatial weight matrix. There are 5 types of local spatial aggregation in the LISA cluster map, namely High-High (HH), Low-Low (LL), Low-High (LH), High-Low (HL), and No Significant (NS). High-High represents the high value of the ERI of both referenced and spatial adjacent cells, Low-Low shows that the ERI value of the referenced and spatial adjacent cells is low, Low-High indicates that the ERI of the referenced cells is low, but the value of the ERI of the spatial adjacent cells is high, High-Low represents the high value of the ERI of the referenced cells, but the low value of the ecological risk index of the adjacent cells [49].

**3. Results**

*3.1. Dynamic Change of LULC from 1990 to 2019*

The spatial distribution of LULC in Xishuangbanna from 1990 to 2019 is shown in Figure 3, and the area and proportion of each LULC type are shown in Table 1. As seen from Table 1, the percentage of forest area was more than 88 % in each period, 89.81% in 1990, 89.24% in 2000, 89.55% in 2010, and 88.39% in 2019, respectively, showing that forest type dominated in Xishuangbanna. Moreover, the percentage of shrub, impervious, water, and others were less than 1% in Xishuangbanna from 1990 to 2019, and the proportion of cropland showed a trend of first increasing, then decreasing and then increasing. Figure 3 shows the cropland was mainly distributed in the fertile soil and water sufficient areas of flat valleys. Shrub, impervious, water, and others were less distributed, and shrubs

were mainly distributed to forest margins lining the southwest of Xishuangbanna. The distribution of impervious is concentrated and shows a trend of continuous expansion from 1990 to 2019.

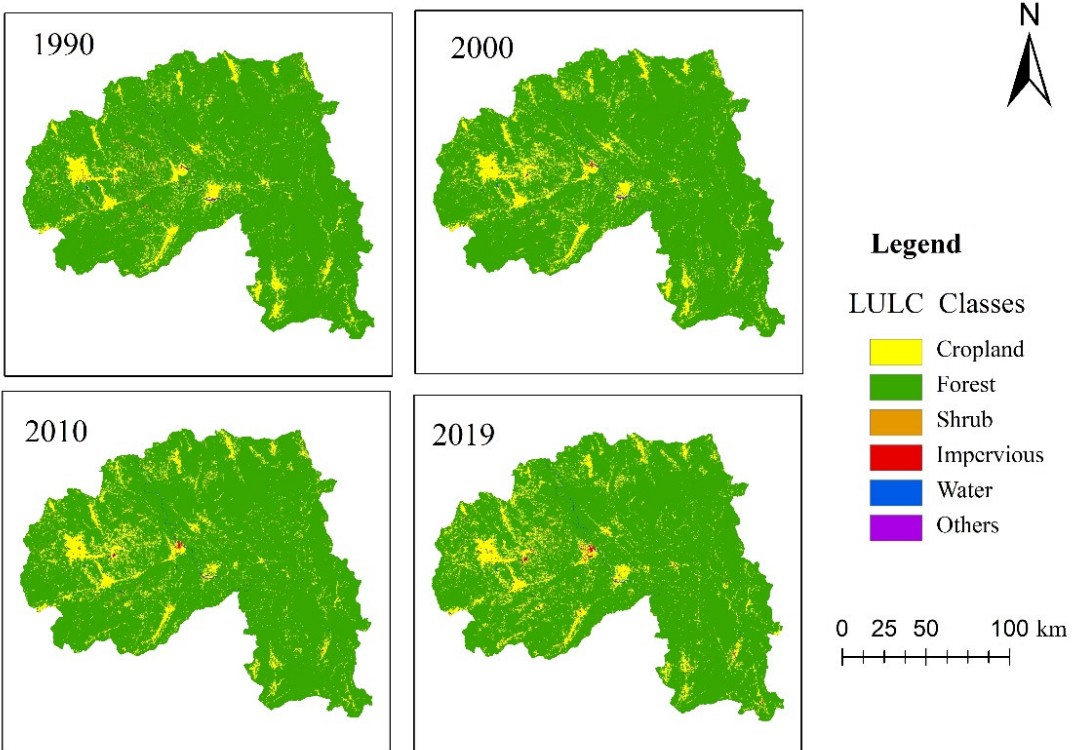

**Figure 3.** Map of land use/land cover for Xishuangbanna Prefecture.

**Table 1.** Area and proportion of each LULC type in Xishuangbanna from 1990 to 2019.

| Year | | Cropland | Forest | Shrub | Impervious | Water | Others |
|------|------|---------|--------|-------|-----------|-------|--------|
| 1990 | Pct. (%) | 9.05 | 89.81 | 0.60 | 0.02 | 0.18 | 0.33 |
| 2000 | Pct. (%) | 10.18 | 89.24 | 0.19 | 0.04 | 0.19 | 0.16 |
| 2010 | Pct. (%) | 9.59 | 89.55 | 0.35 | 0.08 | 0.22 | 0.21 |
| 2019 | Pct. (%) | 10.60 | 88.39 | 0.39 | 0.21 | 0.23 | 0.18 |

Figure 4 shows the changing trend of forest and cropland was the most obvious from 1990 to 2019 in Xishuangbanna, whereas the changing trend of shrub, impervious, water, and other land types was not obvious in the whole period. In addition, the forest was mainly transformed into cropland, and cropland was mainly transformed into the forest shown in Figure 4. The LULC transfer matrix in Xishuangbanna is shown in Table 2. The result shows that the total area of forest, shrubs, and other land types in Xishuangbanna from 1990 to 2019 was 913.35 km$^2$ converted into cropland, whereas cropland transferred 620.26 km$^2$ to another land type. The area of forest-type conversion to other types was 929.55 km$^2$, and 651.51 km$^2$ of another was converted to forest type. In addition, Table 2. Shows that the total area of the impervious and water gradually increased from 1990 to 2019 in Xishuangbanna, with an increase of 34.87 km$^2$ and 9.93 km$^2$, respectively. The cropland area increased by 293.09 km$^2$, forest decreased by 278.04 km$^2$, but the area of shrubs and others reduced by 32.09 km$^2$ and 27.75 km$^2$.

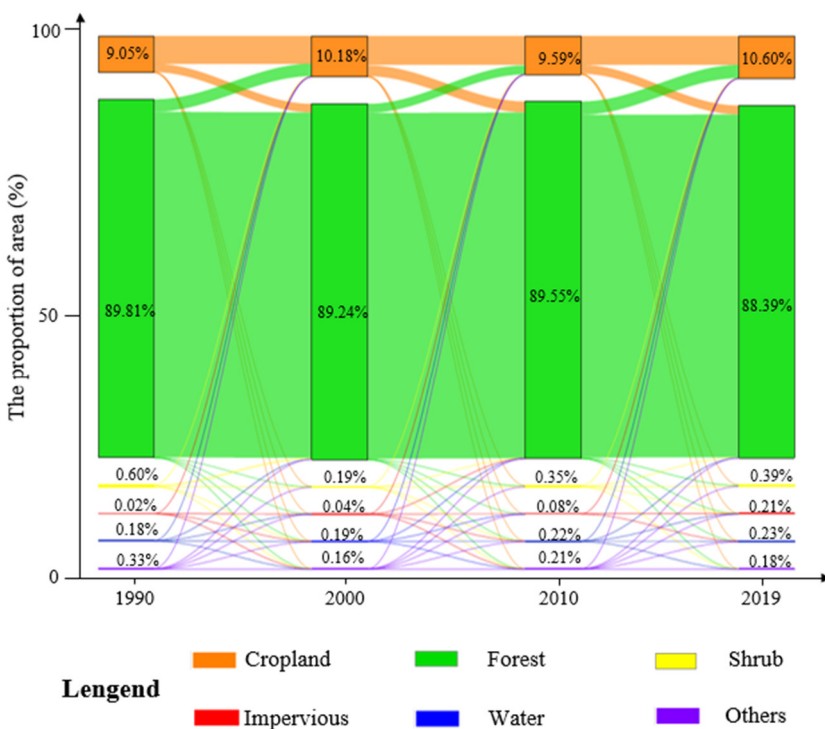

**Figure 4.** Sankey plot of land-use change.

**Table 2.** The area of land use/land cover transfer in Xishuangbanna prefecture from 1990 to 2019.

| Year | Land Type/km² | Cropland | Forest | Shrub | Impervious | Water | Others |
|---|---|---|---|---|---|---|---|
| 1990 to 2000 | Cropland | 1293.46 | 407.79 | 3.10 | 2.97 | 3.75 | 12.17 |
| | Forest | 597.94 | 16,486.31 | 13.43 | 0.02 | 0.11 | 1.42 |
| | Shrub | 25.11 | 61.90 | 18.36 | 0.00 | 8.32 | 0.62 |
| | Impervious | 0.01 | 0.00 | 0.00 | 4.06 | 0.20 | 0.01 |
| | Water | 2.83 | 1.15 | 0.00 | 0.06 | 30.21 | 0.41 |
| | Others | 18.12 | 24.95 | 1.60 | 0.28 | 2.10 | 15.54 |
| 2000 to 2010 | Cropland | 1373.41 | 522.16 | 8.71 | 6.30 | 7.20 | 19.69 |
| | Forest | 434.20 | 16,499.93 | 43.92 | 0.34 | 0.20 | 3.50 |
| | Shrub | 6.60 | 15.36 | 14.02 | 0.00 | 0.00 | 0.51 |
| | Impervious | 0.01 | 0.00 | 0.00 | 6.90 | 0.47 | 0.00 |
| | Water | 2.35 | 1.74 | 0.00 | 0.41 | 31.66 | 0.20 |
| | Others | 7.82 | 2.77 | 0.66 | 1.56 | 1.92 | 15.41 |
| 2010 to 2019 | Cropland | 1351.25 | 427.20 | 8.55 | 15.65 | 4.61 | 17.12 |
| | Forest | 636.03 | 16,357.61 | 44.10 | 1.72 | 0.20 | 2.28 |
| | Shrub | 14.79 | 31.20 | 20.91 | 8.33 | 0.00 | 0.41 |
| | Impervious | 0.01 | 0.00 | 0.00 | 14.90 | 0.62 | 0.00 |
| | Water | 2.74 | 0.31 | 0.00 | 0.12 | 38.25 | 0.04 |
| | Others | 11.48 | 4.88 | 0.33 | 6.75 | 0.90 | 14.98 |
| 1990 to 2019 | Cropland | 1102.95 | 555.70 | 4.45 | 29.85 | 10.57 | 19.69 |
| | Forest | 859.93 | 16,169.67 | 55.97 | 2.45 | 1.10 | 10.10 |
| | Shrub | 28.02 | 66.06 | 11.36 | 0.02 | 0.03 | 0.50 |
| | Impervious | 0.06 | 0.00 | 0.00 | 3.50 | 0.71 | 0.00 |
| | Water | 4.29 | 2.71 | 0.00 | 0.31 | 27.29 | 0.05 |
| | Others | 21.05 | 27.04 | 2.12 | 3.01 | 4.87 | 4.49 |
| | Total | 2016.30 | 16,821.18 | 73.90 | 39.14 | 44.57 | 34.83 |
| | Variation | 293.09 | −278.04 | −32.09 | 34.87 | 9.92 | −27.75 |

*3.2. Spatial-Temporal Changes of Landscape Ecological Risk from 1990 to 2019*

We listed the area and percentages of every ERI grade in Table 3. The result shows that the total proportion of the grade of the Lowest risk was the largest from 1990 to 2019, accounting for 74%, 66%, 59%, and 66%, respectively, and the proportion of Middle-risk grade, Higher risk grade, and Highest risk grade was less than 10% in each period. In addition, the area proportion of Lower risk gradually increased, with 3230.73 km$^2$, 3348.30 km$^2$, 3509.83 km$^2$, and 3718.04 km$^2$ from 1990 to 2019. Table 3 shows that the total proportion of the Lowest landscape ecological risk decreased from 1990 to 2010 and increased from 2010 to 2019. The spatial-temporal distribution of landscape ecological risk in Xishuangbanna is shown by Figure 5, and the lowest risk areas were the most widely distributed in Xishuangbanna, and the areas with Highest and Higher ecological risks were concentrated in the western regions with frequent human activities. Figure 5 shows that the landscape ecological risk of Xishuangbanna first spread and then contracted from 1990 to 2019. Moreover, the Lowest and Lower risk areas were most widely distributed in the Nature Reserve and were covered mainly by a large area of forests. The overall distribution characteristics are higher in the west than in the east of Xishuangbanna.

**Table 3.** Area and percentages of every ERI grade in different years.

| ERI Grade | | Lowest/ (0–0.03) | Lower/ (0.03–0.1) | Middle/ (0.1–0.2) | Higher/ (0.2–0.4) | Highest/ (0.4–1) |
|---|---|---|---|---|---|---|
| 1990 | Area (km$^2$) | 14,139.52 | **3230.73** | 1321.91 | 319.48 | 30.51 |
| | Pct. (%) | **74.27** | 16.97 | 6.94 | 1.68 | 0.16 |
| 2000 | Area (km$^2$) | 12,530.76 | **3348.30** | 1672.80 | 1277.93 | 208.20 |
| | Pct. (%) | **65.82** | 17.59 | 8.79 | 6.71 | 1.09 |
| 2010 | Area (km$^2$) | 11,286.93 | **3509.83** | 1982.41 | 1491.52 | 767.30 |
| | Pct. (%) | **59.29** | 18.44 | 10.41 | 7.83 | 4.03 |
| 2019 | Area (km$^2$) | 12,530.76 | **3718.04** | 1706.90 | 981.78 | 109.49 |
| | Pct. (%) | **65.82** | 19.53 | 8.97 | 5.16 | 0.58 |

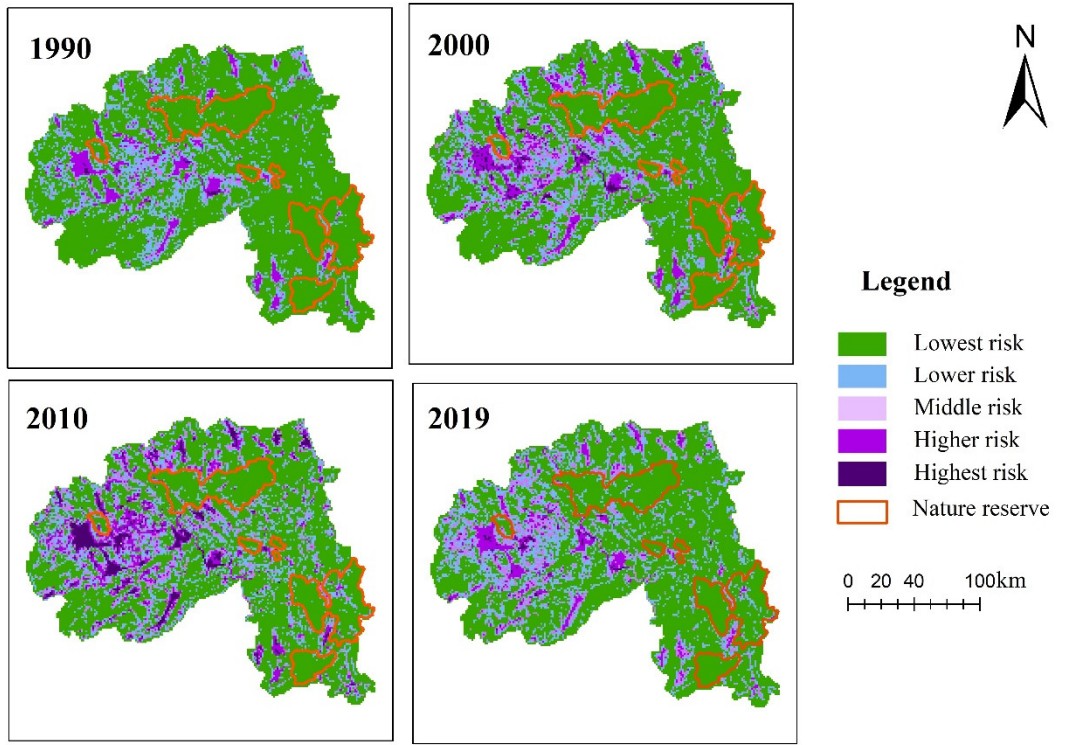

**Figure 5.** Landscape ecological risk level distribution map of Xishuangbanna.

The area and proportion of each ERI level (Lowest, Lower, Middle, Higher, and Highest) were calculated from four landscape ecological risk spatial distribution maps in 1990, 2000, 2010, and 2019, respectively. Figure 6 shows the area distribution of ecological risk levels; the total proportion of the Lowest decreased from 1990 to 2010, but the reverse trend was shown from 2010 to 2019. However, the total percentage of the ERI of Middle, Higher, and Highest increased from 1990 to 2010, but the area proportion of ERI of Middle, Higher, and Highest decreased from 2010 to 2019. The area percentage of ERI of Lower gradually increased during 1990–2019. Therefore, it could be summed up that the landscape ecological risk of the Xishuangbanna firstly increased and then decreased.

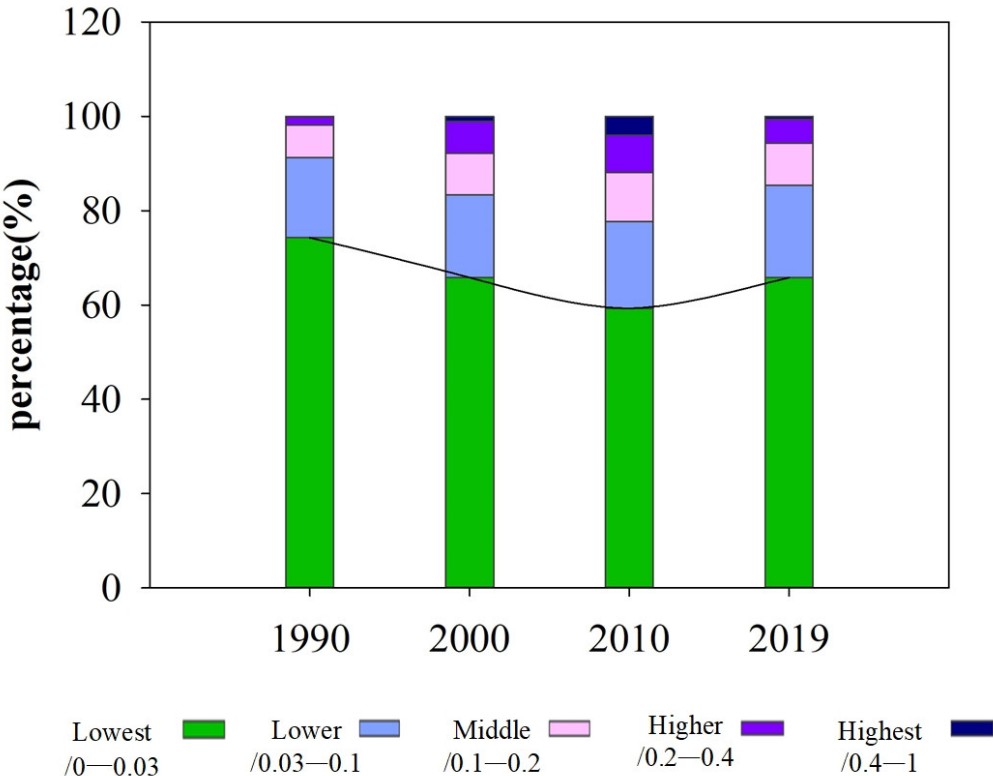

**Figure 6.** Area distribution of ecological risk levels in Xishuangbanna prefecture from 1990 to 2019.

Based on the results of landscape ecological risk level classification in 1990, 2000, 2010, and 2019, we listed the different ERI performances and the spatial-temporal patterns of the landscape ecological risk of the Xishuangbanna in Table 4. The difference performances of the ERI were calculated for three periods by us, 1990–2000, 2000–2010, and 2010–2019, and the results were divided into five levels. They were Obvious Decrease (OD), Slight Decrease (SD), Invariability (IN), Slight Increase (SI), and Obvious Increase (OD), respectively. According to the results, the landscape ecological risk was invariable in most regions in Xishuangbanna, and the percentage of the change level of IN gradually decreased in three periods, accounting for 72.48%, 71.95%, and 69.4%, respectively. The percentage of the change level of OI and OD was under 1%, indicating that the regions in Xishuangbanna with significant changes were few during 1990–2019 in landscape ecological risk. The percentage of the SD gradually increased, with the change percentage of 3.59%, 5.54%, and 24.89% in the three periods, respectively. The change percentage of the SI was 23.73%, 22.12%, and 5.13%, respectively, which gradually decreased from 1990 to 2019, indicating that the ecological environment of Xishuangbanna gradually improved. The increase in the OS in the 1990 to 2010 period was due to human unreasonable land use. So, it is believed that the ERI of the region of interest firstly increased and then decreased.

**Table 4.** Change detection of landscape ecological risk (ERI) level from 1990 to 2019.

| Years | | Obvious Decrease | | Slight Decrease | | Invariability | Slight Increase | | Obvious Increase | |
|---|---|---|---|---|---|---|---|---|---|---|
| | change level | ↓4 | ↓3 | ↓2 | ↓1 | 0 | ↑1 | ↑2 | ↑3 | ↑4 |
| | area/km² | 0.00 | 3.59 | 52.05 | 630.90 | 13,798.15 | 3977.46 | 541.16 | 35.00 | 0.00 |
| 1990 to 2000 | change/km² | 3.59 | | 682.95 | | 13,798.15 | 4518.61 | | 35.00 | |
| | percentage/% | 0.02% | | 3.59% | | 72.48% | 23.73% | | 0.18% | |
| | change level | ↓4 | ↓3 | ↓2 | ↓1 | 0 | ↑1 | ↑2 | ↑3 | ↑4 |
| | area/km² | 0.00 | 17.95 | 116.67 | 938.72 | 13,696.73 | 3714.51 | 497.18 | 51.15 | 5.38 |
| 2000 to 2010 | change/km² | 17.95 | | 1055.39 | | 13,696.73 | 4211.69 | | 56.54 | |
| | percentage/% | 0.09% | | 5.54% | | 71.95% | 22.12% | | 0.30% | |
| | change level | ↓4 | ↓3 | ↓2 | ↓1 | 0 | ↑1 | ↑2 | ↑3 | ↑4 |
| | area/km² | 3.59 | 99.62 | 578.85 | 4160.53 | 13,212.12 | 883.08 | 93.33 | 6.28 | 0.90 |
| 2010 to 2019 | change/km² | 103.20 | | 4739.38 | | 13,212.12 | 976.42 | | 7.18 | |
| | percentage/% | 0.54% | | 24.89% | | 69.40% | 5.13% | | 0.04% | |

Change level included ↑4, ↑3, ↑2, ↑1, 0, ↓1, ↓2, ↓3, and ↓4. ↑4, ↑3, ↑2, and ↑1 indicated that landscape ecological risk increase level 4, 3, 2, and 1, respectively. Additionally, 0 represented the landscape ecological risk level that remained constant, but ↓1, ↓2, ↓3, and ↓4 represented landscape ecological risk which decreased level 4, 3, 2, and 1, respectively.

### 3.3. Spatial Autocorrelation Analysis of Landscape Ecological Risk

Based on the value of ecological risk assessment cells mentioned above, the Moran's I and LISA were used to analyze the spatial autocorrelation of the ERI in Xishuangbanna. Figure 7 shows the Moran's I scatter plot of the ERI included in 1990, 2000, 2010, and 2019, and the points were mainly distributed in quadrants one and three, indicating that the landscape ecological risk in the study area has a strong positive spatial correlation. The values of Moran's I reached 0.655, 0.650, 0.652, and 0.633 in 1990, 2000, 2010, and 2019, respectively, which shows that the spatial distribution of landscape ecological risk in Xishuangbanna shows clustering rather than randomness. Moreover, the values of Moran's I shows a gradually increasing trend from 1990 to 2010, and then a decreased trend from 2010 to 2019, which is consistent with the landscape ecological risk levels (Figure 6).

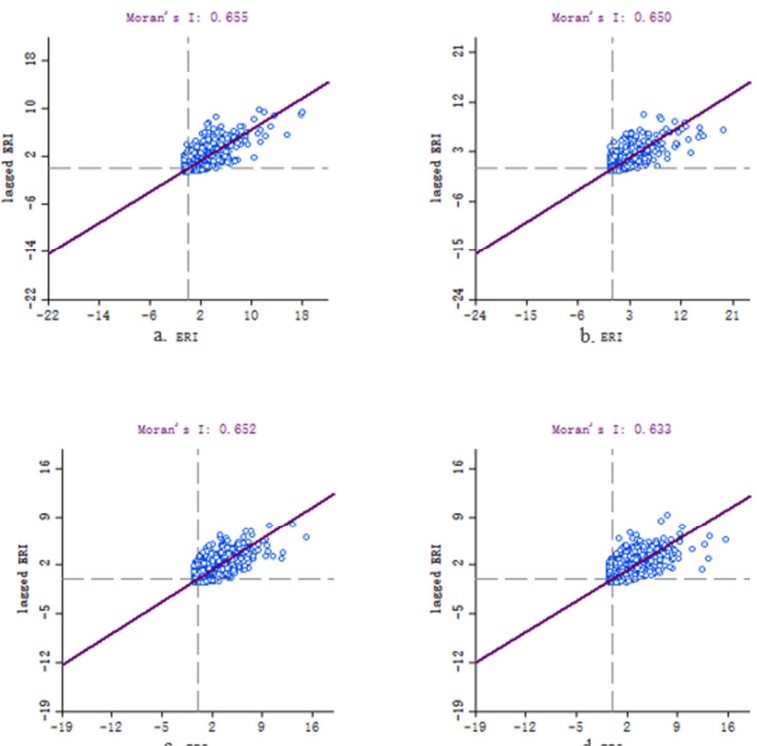

**Figure 7.** Moran's I scatter plots of the ERI in Xishuangbanna prefecture in (**a**) 1990, (**b**) 2000, (**c**) 2010, and (**d**) 2019.

To understand the spatial-temporal distribution of the landscape ecological risk, the LISA cluster map was used to analyze the local spatial correlation pattern of the ERI. The LISA cluster map (Figure 8) shows that the Not Significant clustering area was mainly distributed in the areas with fewer changes of LULC, and the High-High (HH) clustering area was located in the west with intensive human activities of Xishuangbanna, and the distribution was similar to the ecological risk level change shown in Figure 5, indicating the high landscape ecological risk and a worse ecological environment. The Low-High (LH) clustering area is scattered and distributed around the HH clustering area in Figure 8. The clustering area of HH increased gradually and spread to the east from 1990 to 2019, which reflect that the threats to Xishuangbanna's ecological environment increased.

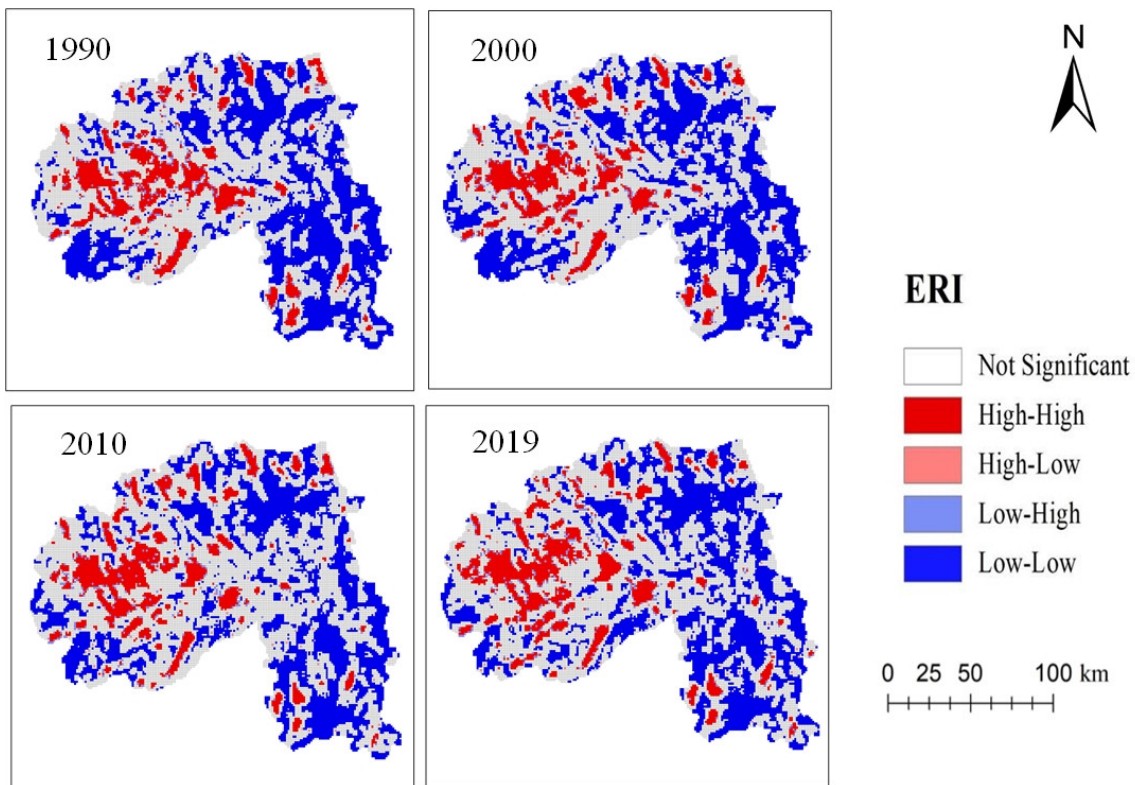

**Figure 8.** The LISA maps of local spatial auto-correlation in 1990, 2000, 2010, and 2019.

## 4. Discussion

### 4.1. The Effect of LULC on Landscape Ecological Risk

The policy is a factor affecting the change of ecological risk. For example, China's government has come up with a series of conservation programs in the National nature reserve, which has improved the eco-environmental quality nowadays [15]. A large number of forests and cropland have been converted to tea plantations and rubber plantations in Xishuangbanna due to the rubber expansion, the land-contracted responsibility system policy implemented, and the economic incentives policy [3,50]. Therefore, to protect tropical forest ecosystems and maintain the sustainability of the ecosystem, Xishuangbanna established national, large integrated nature reserves in 1958 [51], developing the pilot program of returning rubber plantations to rainforests, and carrying out the project of ecological protection and restoration. This has helped to improve public awareness of rainforest protection and ecologically sustainable development [52]. Meanwhile, we found that the LULC changes in Xishuangbanna from 1990 to 2019 were reduced in forests and increased in cropland and shrubs. With the process of urbanization, the expansion of impervious has led to deforestation and degradation in the urban fringe, which may also lead to an increasing risk of landscape ecology [4].

The expansion of rural settlements will lead to changes in ecological risks because human activities have more impact on the sustainable development of an ecosystem [53]. Consequently, the higher ERI area and the highest ERI area in Xishuangbanna were mainly distributed in the areas with cropland and impervious. According to statistics, the population of Xishuangbanna was 7.96 million, 9.39 million, 11.25 million, and 11.96 million in 1990, 2000, 2010, and 2019, respectively, showing an increasing trend [51,54]. Theoretically, the impervious area will increase with population growth, and the ecological environment in the impervious area is fragile.

The landscape ecological risk has increased from 1990 to 2010. The high ERI is mainly distributed in the western part of the study area and the low ERI is mainly distributed in the eastern and southwestern area, which has benefited from the national eco-environmental protection policy and the strengthening of the nature reserve protection [55]. However, the areas with a medium and high risk in Xishuangbanna decreased by 1143.06 km$^2$ from 2010 to 2019, the serious situation of high-landscape ecological risk has been greatly alleviated, and the Lower and Lowest risk areas has gradually increased.

The landscape ecological risk of Xishuangbanna is lower than that of other areas because the national nature reserve has strong conservation policies. Furthermore, the proportion of the Lowest landscape ecological risk area was the largest. In addition, the changing trend of the landscape ecological risk around nature reserves was the same, in that it firstly increased and then decreased. The Manggao Nature Reserve was higher than other nature reserves in landscape ecological risk from 1990 to 2019. Although the landscape ecological risk began to reduce after 2010, the land use and environmental issues in Manggao Nature Reserve require more attention.

*4.2. Strengths and Limitations*

To provide a decision-making basis for formulating scientific management of land resources and sustainable development of the ecological environment, the spatial-temporal pattern analysis of landscape ecological risk can monitor and identify the outstanding problems and potential risks [10]. The landscape ecological risk assessment method based on LULC is the most extensively applied method in the region of ecological risk evaluation; the spatiotemporal expression of ecological risk is realized based on multi-scale without a large number of field observations [15]. The strengths of this study are that we analyzed the spatial and temporal changes of landscape ecological risk based on the GEE cloud platform in this study. The GEE can analyze the dynamic change of LULC efficiently and calculate the value of landscape ecological risk quickly and provide convenience for spatial autocorrelation analysis of ecological risk, compared to the traditional software, such as ArcGIS and Fragstats.

Although this approach has shown its effectiveness in spatiotemporal variability in landscape ecological risk assessment, there are some notable uncertainties in this method. First of all, the result of the LULC affects the result of the landscape ecological risk assessment, and the LULC errors may lead to uncertainty in the ERA [28]. The overall accuracy of CLCD reached 79.3% from 1990 to 2019 in the study area, which is reliable for our research. However, LULC classification errors are inevitable, so improving the high accuracy of LULC data is a meaningful direction to change the uncertainty of the ERA in the future. Secondly, the difference of low-risk values in Xishuangbanna is small, so the scatter plots appeared to show a stacking phenomenon when we made the Moran's I scatter plots of the ERI (Figure 7). Due to this, the landscape ERA method only considers the area ratio of different landscape types and lacks the ecological connotation of ecological risk. To calculate the ERA based on the source-sink landscape model is an important trend in the future, which could explore whether the landscape types can improve or impede the development of the ecological processes. In addition, the comprehensive Land-Use Landscape Evaluation Method, the Ecological pressure index system, and the driving elements of spatial and temporal changes in ecological risk should be considered in further studies, such as water pollution data, soil quality, and climate change [11,15,29]. In despite of the

uncertainties mentioned above, this study conducted an ecological risk assessment based on the most popular methods, and verified the impact of LULC changes on landscape ecological risk, which can provide a valuable reference for the protection and balance of ecology in Xishuangbanna [56].

## 5. Conclusions

Based on the GEE platform and the China Land Cover Data, we used LULC to conduct spatio-temporal pattern analysis and dynamic change discussion on the landscape ecological risk in the Xishuangbanna region over the past 30 years. It is important to guide rational land use and sustainable development of the ecological environment. The results showed that the dominant LULC type was always Forest in Xishuangbanna, counting for more than 88%. The most outstanding diversion characteristic of LULC was the increase in cropland area and the decrease in forest area with the population and society's economic development from 1990 to 2019. In other words, the landscape ecological risk assessment showed an increasing trend from 1990 to 2010, but it has decreased during 2010 to 2019 because the awareness of ecological environmental protection has increased. The Higher and Highest risks were mainly distributed in the areas with cropland and impervious of LULC, and the Lower and Lowest risk covered the forest regions. Moreover, the changes in ecological risk of the national nature reserve indicated that scientific environmental protection policies played an important role in the sustainable development of the eco-environment. The Moran's I value of the ERI was 0.655, 0.650, 0.652, and 0.633 in 1990, 2000, 2010, and 2019, respectively, showing that the landscape ecological risks in the study area were distributed centrally rather than randomly, and have a strongly positive spatial correlation. The spatial clustering of ecological risk was gradually shortened during 1990 to 2000 and 2010 to 2019, and it was slightly enhanced from 2000 to 2010. The distributions of HH and LL clustered, and the H-H region was mainly distributed in the western areas of the Xishuangbanna, with dense population distribution and frequent human activities. L-L was mainly located in the eastern part of four sub-protected areas in Xishuangbanna National Nature Reserve. In conclusion, the landscape ecological risk first increased and then decreased in Xishuangbanna, so the quality of the ecological conditions improved from 2010 to 2019, which indicated that the land change caused by various policies obviously affects the landscape ecological risk. This study provided a reference for ecological risks assessment and the sustainable development of land resources and ecology.

**Author Contributions:** Conceptualization, writing—original draft, Y.L.; methodology, writing—review and editing, W.X.; data curation, formal analysis, Z.H.; software, visualization, L.W.; software, validation, G.O.; investigation, N.L. All authors have read and agreed to the published version of the manuscript.

**Funding:** This research was supported in part by research grants from the National Natural Science Foundation of China (32060320, 31860181, 32160368); Research Foundation for Basic Research of Yunnan Province (202101AT070039); "Ten Thousand Talents Program" Special Project for Young Top-notch Talents of Yunnan Province (YNWR-QNBJ-2020047); Joint Special Project for Agriculture of Yunnan Province, China (202101BD070001-066).

**Acknowledgments:** We thank the anonymous reviewers for their constructive comments on the earlier version of the manuscript.

**Conflicts of Interest:** The authors declare no conflict of interest.

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
