# Peer review of "Assessment of Spatial-Temporal Changes of Landscape Ecological Risk in Xishuangbanna, China from 1990 to 2019"

_sustainability, doi:10.3390/su141710645_

Round 1

Reviewer 1 Report

An interesting and important article providing commentary on the change landscape in this industrialised area. Before publication I recommend a proof read as some sentences read more as statements - too frequently, for example some sentences need a verb or noun to focus the attention of the content. I appreciate this article may be in a second language to that of the authors, but to credit them and their research, I'd recommend light copy-editing before publishing this important work.

Author Response

The response to your review comments is (Reviewer # 1:) in the cover letter,

Reviewer 2 Report

Dear Authors,

The topic of your study is interesting.

However, at its current state you must reduce the percentage of plagiarism.

Many parts of the introduction and methods is repeated from other studies that you have referred. Better paraphrasing must be performed.

Furthermore, the manuscript must be proofread as there are many typo errors. Some sentences are too long, they could be split into shorter sentences.

 GEE is abruptly introduced in the introduction part. May be some need for large scale data utility could be mentioned before justifying the need for GEE platform.

 Paragraphs in the Introduction part must be better integrated. The transition must be smoother.

Finally, your manuscript is relatively technical bringing an assessment procedure. However, you must tightly discuss the topic in relation to sustainable development. This will make your manuscript more relevant to the journal scope.

Thank you for your understanding!

Reviewer

Author Response

The response to your review comments is (Reviewer # 2:) in the cover letter,

Reviewer 3 Report

Dear Authors,

I have gone through the submission about spatial-temporal changes in landscape ecological risk in Xishuangbanna, China. The data is well presented and discussed in detail. Besides, I found some typos and missing letters in the text as I indicated in the PDF file. There are some inconsistencies in font type and sizes. Besides the resolution of the figs are very low. I assume that this is related to reducing the size of the pdf. If not, please increase the resolution of all figs.  In my opinion, the ms should be worth publishing after minor revision. Best regards,

Author Response

The response to your review comments is (Reviewer # 3:) in the cover letter,

Reviewer 4 Report

I submit the following comment to both the editor and the authors. I am very interested in the research topic. It is definitely a contribution to world knowledge. In particular, the in-text citations need to be adjusted as they are regionally focused. It is indeed necessary to link the research to world knowledge, both in the citations and in the description of the topics. The text is full of minor typos or excess spaces between words and the like. It is necessary to shorten the abstract, and there are statements at the end about the impact of regulation on change, but how has it actually manifested itself? I see some references in the conclusions for the first time. They need to be properly linked to the results and especially the discussion.

Specific comments with lines:

39: from -space - 1996

Introduction-first paragraph: I suggest that all changes in percentages be put in a summary table and added to the introduction.

73: However

138: based

139: 1 km x 1 km-How does the result change with a different cell size?

171-173: The categories overlap. it is necessary to express the categories better.

Table 2 and 3: I suggest bolding in the tables what you describe in the results. It is difficult to navigate the tables.

215: Table 2

220: from 2000

221: increasing3.2 Spatial-temporal- I do not understand this statement.

225: 59% area etc.

225: the highest

248: Error! Reference source not found.. ????

272: 8),the Not significant clustering area - ????

284: The policy is a factor affecting the change of ecological risk. - It would be good to find some analogies in the world. This is a very elementary statement.

328-329: eco-environment 328 protection year by year - First mention. Link to results and discussion.

Author Response

The response to your review comments is (Reviewer # 4:) in the cover letter,
